# Task-Agnostic Continual Reinforcement Learning: Gaining Insights and Overcoming Challenges

**Massimo Caccia**[†][*]
Mila - Quebec AI Institute
Université de Montréal

**Jonas Mueller**[†]
Cleanlab

**Taesup Kim**[†]
Seoul National University

**Laurent Charlin**
Mila - Quebec AI Institute
HEC Montréal
Canada CIFAR AI Chair

**Rasool Fakoor**
Amazon Web Services

## Abstract

We study methods for task-agnostic continual reinforcement learning (TACRL). TACRL combines the difficulties of partially observable RL (due to task agnosticism) and the challenges of continual learning (CL), which involves learning on a non-stationary sequence of tasks. As such, TACRL is important in real-world applications where agents must continuously adapt to changing environments. Our focus is on a previously unexplored and straightforward baseline for TACRL called replay-based recurrent RL (3RL). This approach augments an RL algorithm with recurrent mechanisms to mitigate partial observability and experience replay mechanisms to prevent catastrophic forgetting in CL. We pose a counterintuitive hypothesis that 3RL could outperform its soft upper bounds prescribed by previous literature: multi-task learning (MTL) methods that do not have to deal with non-stationary data distributions, as well as task-aware methods that can operate under full observability. Specifically, we believe that the challenges that arise in certain training regimes could be best overcome by 3RL enabled by its ability to perform *fast adaptation*, compared to task-aware approaches, which focus on task memorization. We extensively test our hypothesis by performing a large number of experiments on synthetic data as well as continuous-action multi-task and continual learning benchmarks where our results provide strong evidence that validates our hypothesis.

## 1 Introduction

Continual learning (CL) creates models and agents that can learn from a sequence of tasks. Continual learning agents aim at solving multiple tasks as well adapt to new tasks without forgetting the previous one(s), a major limitation of standard deep learning agents (French, 1999; Thrun & Mitchell, 1995; McCloskey & Cohen, 1989; Lesort et al., 2020). In many studies, the performance of CL agents is compared against *multi-task* (MTL) agents who are trained in all available tasks jointly. During learning and evaluation, these multi-task agents are typically provided with the identity of the current task (e.g. each datum is coupled with its task ID) making them *task-aware*. The performance of multi-task agents provides a soft upper bound on the performance of continual learning agents that are bound to learn tasks sequentially and so can suffer from *catastrophic forgetting* (McCloskey & Cohen, 1989). Moreover, continual learning agents are often trained without task IDs, a challenging setting motivated by practical constraints and known as *task-agnostic* CL (Zeno et al., 2019; He et al., 2019; Caccia et al., 2020; Berseth et al., 2021).

In this work, our aim is to understand the factors that explain the difference in performance between continual-learning methods and their multi-task counterparts. We hypothesize that both task-agnosticity and continual learning may provide advantages when learning from limited data and computation or in settings where the dimensionality of (the observation space of) each task and the number of tasks to solve are high. Our reasoning is as follows. Task-agnostic methods may learn the ability to adapt more quickly to novel environments that are similar to previous environments. In comparison, task-aware methods may learn to memorize environments and as a result may require more data or computations to adapt to other environments. We further hypothesize that this faster adaptation may be particularly beneficial in continual-learning settings, where it can help combat the impact of catastrophic forgetting.

---

[*] corresponding author: `massimo.p.caccia@gmail.com`    [†]work done while at Amazon Web Services

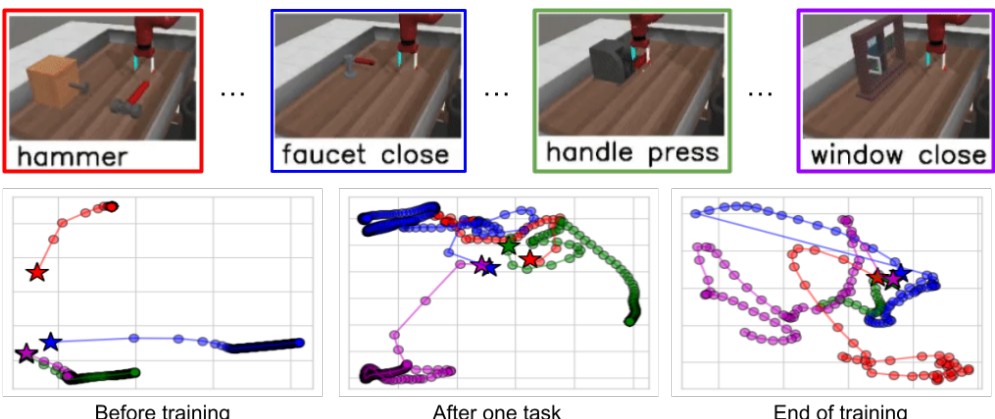

Figure 1: **The Continual-World[2] benchmark (top) as well as evolving RNN representations (bottom).** Continual World consists of 10 robotic manipulation environments (four of which are shown above) within the same state space and built on a common reward structure composed of shared components, i.e, reaching, grasping, placing and pushing. We explore the task-agnostic setting in which agents need at least part of a trajectory (rather than a single state) for task identification. We performed a PCA analysis of 3RL's RNN representations at different stages of training. One episode is shown per task, and the initial task representation (at $t = 0$) is represented by a star. As training progresses, the model learns i) a task-invariant initialization, drawing the initial states closer together, and ii) richer, more diverse representations. Furthermore, the representations constantly evolve throughout the episodes, suggesting the RNN performs more than just task inference: it provides useful local information to the policy and critic.

To evaluate the above hypotheses, we instantiate several task-aware and task-agnostic methods. For the task-agnostic continual-learning agents, we add a recurrent memory and the capability of replaying the trajectory of previous tasks (Rolnick et al., 2019). We refer to this methodology as *replay-based recurrent reinforcement learning* (3RL) (see the bottom of Figure 1 for a visualization of its representations).

We evaluate these methods using two benchmarks. The first benchmark is a synthetic task in which agents learn to maximize a quadratic function. The dimensions of the observation and action spaces are controllable. Our results show that 3RL outperforms baseline methods, especially with respect to robustness, as we decrease data and compute resources or increase the dimensionality of the problem. This improvement might be due to a reduction in gradient conflict. Next, we evaluate our algorithm on the Meta-World Yu et al. (2019) benchmark, which consists of 50 distinct manipulation tasks. Our results show that 3RL is the superior approach for CRL and is able to reach its MTRL soft-upper bound, a remarkable achievement that no other CRL has accomplished to the best of our knowledge. Our results the code to reproduce them are publically available[1].

## 2   BACKGROUND & TASK-AGNOSTIC CONTINUAL REINFORCEMENT LEARNING (TACRL)

Here, we define TACRL, and contrast it against multi-task RL as well as task-aware settings.

**MDP.**   The RL problem is often formulated using a Markov decision process (MDP) (Puterman, 1994). An MDP is defined by the five-tuple $\langle \mathcal{S}, \mathcal{A}, \mathcal{T}, r, \gamma \rangle$ with $\mathcal{S}$ the state space, $\mathcal{A}$ the action space, $\mathcal{T}(s'|s,a)$ the transition probabilities, $r(s,a) \in \mathbb{R}$ or equivalently $r_i$ the reward obtained by taking action $a \in \mathcal{A}$ in state $s \in \mathcal{S}$, and $\gamma \in [0,1)$ a scalar constant that discounts future rewards. In RL, the transition probabilities and the rewards are typically unknown and the objective is to learn a policy, $\pi(a|s)$ that maximizes the sum of discounted rewards $\mathcal{R}_t^\pi = \sum_{i=t}^\infty \gamma^{i-t} r_i$ generated by taking a series of actions $a_t \sim \pi(\cdot|s_t)$. The Q-value $Q^\pi(s,a)$ corresponding to a policy $\pi$, is defined as the expected return starting at state $s$, taking $a$, and acting according to $\pi$ thereafter: $Q^\pi(s,a) = \mathbb{E}_\pi \left[ \sum_{t=0}^\infty \gamma^t r_t \right] = r(s,a) + \gamma \mathbb{E}_{s',a'} \left[ Q^\pi(s',a') \right]$.

---

[1] https://github.com/amazon-science/replay-based-recurrent-rl

[2] The figures depict the rendering of Meta-World, and not what the agent observes. The agent's observation space is composed of object, targets, and gripper position. Because of the randomness of those positions, the agents need more than one observation to properly infer the hidden state.

**POMDP.** In most real-world applications, if not all, the full information about an environment or a task is not always available to the agent due to various factors such as limited and/or noisy sensors, different states with identical observations, occluded objects, etc. (Littman et al., 1995; Fakoor & Huber, 2012). For this class of problems in which environment states are not fully observable by the agent, partially-observable Markov decision processes (POMDPs) Kaelbling et al. (1998) are used to model the problem. A POMDP is defined by a seven-tuple $\langle \mathcal{S}, \mathcal{A}, \mathcal{T}, \mathcal{X}, \mathcal{O}, r, \gamma \rangle$ that can be interpreted as an MDP augmented with an observation space $\mathcal{X}$ and an observation-emission function $\mathcal{O}(x'|s)$. In a POMDP, an agent cannot directly infer the current state of the environment $s_t$ from the current observation $x_t$. We split the state space into two distinct parts: the one that is observable $x_t$, which we refer to as $s_t^o$, and the remainder as the hidden state $s_t^h$, similarly to (Ni et al., 2021). To infer the correct hidden state, the agent has to take its history into account: the policy thus becomes $\pi(a_t|s_{1:t}^o, a_{1:t-1}, r_{1:t-1})$. An obvious choice to parameterize such a policy is with a recurrent neural network (Lin & Mitchell, 1993; Whitehead & Lin, 1995; Bakker, 2001; Fakoor et al., 2020b; Ni et al., 2021), as described in Section 3. Like MDPs, the objective in POMDP is to learn a policy that maximizes the expected return $\mathbb{E}_{s^h}\left[\mathbb{E}_{\pi}\left[\sum_{t=0}^{\infty} \gamma^t r_t\right]|s^h\right]$.

**Task-agnostic Continual Reinforcement Learning (TACRL).** TACRL agents operate in a POMDP special case, explained next, designed to study the *catastrophic forgetting* that plagues neural networks (McCloskey & Cohen, 1989) when they learn on non-stationary data distributions, as well as *forward transfer* (Wolczyk et al., 2021a), i.e., a method's ability to leverage previously acquired knowledge to improve the learning of new tasks (Lopez-Paz & Ranzato, 2017). First, TACRL's environments assume that the agent does not have a causal effect on $s^h$. This assumption increases the tractability of the problem. It is referred to as a hidden-mode MDP (HM-MDP) (Choi et al., 2000). Table 1 provides its mathematical description.

The following assumptions help narrow down the forgetting problem and knowledge accumulation abilities of neural networks. TACRL's assumes that $s^h$ follows a non-backtracking chain. Specifically, the hidden states are locally stationary and are never revisited. Finally, TACRL's canonical evaluation reports the anytime performance of the methods on all tasks, which we will refer to as *global return*. In this manner, we can tell precisely which algorithm has accumulated the most knowledge about all hidden states at the end of its *life*. In TACRL, the hidden states can be changed into a single categorical variable often referred to as *context*, but more importantly in CRL literature, it represents a *task*. As each context can be reformulated as a specific MDP, we treat tasks and MDPs as interchangeable.

**Awareness of the Task Being Faced.** In practical scenarios, deployed agents cannot always assume full observability, i.e. access to a *task label* or *ID* indicating which task they are solving or analogously which state they are in. They might not even have the luxury of being "told" when the task changes (unobserved task boundary) in which case agents might have to infer it from data. This setting is known as *task agnosticism* (Zeno et al., 2019). Although impractical, CL research often uses *task-aware* methods, which observe the task label, as soft upper bounds for task-agnostic methods (van de Ven & Tolias, 2019; Zeno et al., 2019). Augmented with task labels, the POMDP becomes fully observable, in other words, it is an MDP.

**Multi-task Learning (MTL).** For neural network agents, catastrophic forgetting results from the stationary-data-distribution assumption of stochastic gradient descent being violated. As a result, the network parameters become specific to data from the most recent task. Thus it is generally preferable to train on data from all tasks jointly as in MTL (Zhang & Yang, 2021). However this may not be possible in many settings, and CL is viewed as a more broadly applicable methodology that is expected to perform worse than MTL (Rolnick et al., 2019; Chaudhry et al., 2018).

For RL specifically, multi-task RL (MTRL) often refers to scenarios with families of similar tasks (i.e. MDPs) where the goal is to learn a policy (which can be contextualized on each task's ID) that maximizes returns across *all* the tasks (Yang et al., 2020; Calandriello et al., 2014; Kirkpatrick et al., 2017a). While seemingly similar to CRL, the key difference is that MTRL assumes data from all tasks are readily available during training and each task can be visited as often as needed. These are often impractical requirements, which CRL methods are not limited by. Table 1 summarizes the settings we have discussed in this section. Refer to App. A for further details and related settings.

| | T | $\pi$ | Objective | Evaluation |
|---|---|---|---|---|
| MDP (Sutton & Barto, 2018) | $p(s_{t+1}\|s_t,a_t)$ | $\pi(a_t\|s_t)$ | $\mathbb{E}_\pi\big[\sum_{t=0}^\infty \gamma^t r_t\big]$ | same as Objective |
| POMDP (Kaelbling et al., 1998) | $p(s_{t+1}^h,s_{t+1}^o\|s_t^h,s_t^o,a_t)$ | $\pi(a_t\|s_{1:t}^o,a_{1:t-1},r_{1:t-1})$ | $\mathbb{E}_{s^h}\big[\mathbb{E}_\pi[\sum_{t=0}^\infty \gamma^t r_t]\|s^h\big]$ | same as Objective |
| HM-MDP (Choi et al., 2000) | $p(s_{t+1}^o\|s_{t+1}^h,s_t^o,a_t)p(s_{t+1}^h\|s_t^h)$ | $\pi(a_t\|s_{1:t}^o,a_{1:t-1},r_{1:t-1})$ | $\mathbb{E}_{s^h}\big[\mathbb{E}_\pi[\sum_{t=0}^\infty \gamma^t r_t]\|s^h\big]$ | same as Objective |
| **Task-agnostic CRL** | $p(s_{t+1}^o\|s_t^h,s_t^o,a_t)p(s_{t+1}^h\|s_t^h)$ | $\pi(a_t\|s_{1:t}^o,a_{1:t-1},r_{1:t-1})$ | $\mathbb{E}_{s^h}\big[\mathbb{E}_\pi[\sum_{t=0}^\infty \gamma^t r_t]\|s^h\big]$ | $\mathbb{E}_{\tilde{s}^h}\big[\mathbb{E}_\pi[\sum_{t=0}^\infty \gamma^t r_t]\|s^h\big]$ |
| Task-Aware CRL | $p(s_{t+1}^o\|s_t^h,s_t^o,a_t)p(s_{t+1}^h\|s_t^h)$ | $\pi(a_t\|s_t^h,s_t^o)$ | $\mathbb{E}_{s^h}\big[\mathbb{E}_\pi[\sum_{t=0}^\infty \gamma^t r_t]\|s^h\big]$ | $\mathbb{E}_{\tilde{s}^h}\big[\mathbb{E}_\pi[\sum_{t=0}^\infty \gamma^t r_t]\|s^h\big]$ |
| Multi-task RL | $p(s_{t+1}^o\|s_t^h,s_t^o,a_t)p(s_{t+1}^h)$ | $\pi(a_t\|s_t^h,s_t^o)$ | $\mathbb{E}_{\tilde{s}^h}\big[\mathbb{E}_\pi[\sum_{t=0}^\infty \gamma^t r_t]\|s^h\big]$ | same as Objective |

Table 1: Summarizing table of the settings relevant to TACRL. For readability purposes, $\tilde{s}^h$ denotes the stationary distribution of $s^h$. The blue colorization highlights the changes occurring from one setting to the next.

## 3 METHODS & HYPOTHESES

In this section, we detail the base algorithm and different model architectures used for assembling different continual and multi-task learning baselines. We also put forth some hypotheses that pertain to the aptitude of diverse modeling approaches to exhibit superior or inferior performance across varied learning scenarios.

### 3.1 ALGORITHMS

We use off-policy RL approaches which have two advantages for (task-agnostic) continual learning. First, they are more sample efficient than online-policy ones (Haarnoja et al., 2018a; Fakoor et al., 2020a). Learning from lower-data regimes is preferable for CRL since it is typical for agents to only spend short amounts of time in each task and for tasks to only be seen once. Second, task-agnostic CRL most likely requires some sort of replay function (Traoré et al., 2019; Lesort et al., 2019). This is in contrast to task-aware methods which can, at the expense of computational efficiency, *freeze-and-grow*, e.g. PackNet (Mallya & Lazebnik, 2018), incur no forgetting. Off-policy methods, by decoupling the learning policy from the acting policy, support the replaying of past data. In short, off-policy learning is the approach of choice in CRL.[3]

**Base algorithm.** We utilize Soft Actor-Critic (SAC) (Haarnoja et al., 2018b) as the base algorithm in this paper. SAC is an off-policy actor-critic method for continuous control, where it learns a stochastic policy that maximizes the expected return while also encouraging the policy to contain some randomness. It uses an actor/policy network $\pi_\phi$ and critic/Q network $Q_\theta$, which are parameterized by $\phi$ and $\theta$, respectively. Refer to App. B for more details.

### 3.2 MODELS

We explore different architectures for MTL and CL in both task-aware and task-agnostic settings.

**Task ID modeling (TaskID).** We assume that a model such as SAC can become task adaptive by providing task information to the networks. Task information such as task ID (e.g. one-hot representation), can be fed into the critics and actor networks as additional input: $Q_\theta(s, a, \tau)$ and $\pi_\phi(a|s, \tau)$ where $\tau$ is the task ID. We refer to this baseline as Task ID modeling (TaskID). This method is applicable in both multi-task learning and continual learning.

**Multi-head modeling (MH).** For multi-task learning (which is always task-aware), the standard SAC is typically extended to have multiple heads (Yang et al., 2020; Yu et al., 2019; Wolczyk et al., 2021b; Yu et al., 2020), where each head is responsible for a single distinctive task, i.e. $Q_\Theta = \{Q_{\theta_k}\}_k^K$ and $\pi_\Phi = \{\pi_{\phi_k}\}_k^K$ with $K$ the total number of tasks. MH is also applicable to all reinforcement learning algorithms. That way, the networks can be split into 2 parts: (1) a shared state representation network (feature extractor) and (2) multiple prediction networks (heads). This architecture can also be used for task-aware CL, where a new head is newly attached (initialized) when an unseen task is encountered during learning.

We also use this architecture in the task-agnostic setting for both MTL and CL. Specifically, the number of heads is fixed a priori (we fix it to the number of total tasks) and the most confident actor head, w.r.t. the entropy of the policy

---

[3]Note that our findings are not limited to off-policy methods, in fact, our 3RL model can be extended to any on-policy method as long as it utilizes a replay buffer (Fakoor et al., 2020a). Having the capability to support a replay buffer is more important than being on-policy or off-policy.

and most optimistic critic head are chosen. Task-agnostic multi-head (TAMH) can help us fraction the potential MH gains over the base algorithm: if MH and TAMH can improve performance, some of MH gains can be explained by its extra capacity instead of by the additional task information.

**Task-agnostic recurrent modeling (RNN).** Recurrent neural networks are capable of encoding the history of past data (Lin & Mitchell, 1993; Whitehead & Lin, 1995; Bakker, 2001; Fakoor et al., 2020b; Ni et al., 2021). Their encoding can implicitly identify different tasks (or MDPs). Thus, we introduce RNNs as a history encoder where the history is defined by $\{(s_i, a_i, r_i)\}_i^N$ and we utilize the hidden states $z$ as additional input data for the actor $\pi_\phi(a|s,z)$ and critic $Q_\theta(s, a, z)$ networks. This allows us to train models without any explicit task information, and therefore we use this modeling, especially for task-agnostic continual learning. More RNN details are provided later in the next subsection.

There is a fundamental distinction between two primary approaches: task-aware and task-agnostic. Although both aim to address the challenge of solving multiple tasks, they differ in their strategies for achieving this objective. Task-aware approaches seek to memorize the solution for each task. Essentially, they attempt to learn and remember the optimal policy for all tasks by sharing parameters in a single neural network. Task-agnostic approaches, on the other hand, aim to achieve the same goal without complete memorization.

Instead, they may focus on learning a general solution that can perform *fast adaptation* (Finn et al., 2017) to previous and new tasks via the task-inference module—typically an RNN—as demonstrated in Ni et al. (2021). We can illustrate the distinction using the example of a massaging robot. A task-aware robot would aim to learn and remember the best massaging approach for each individual client, while a task-agnostic robot would instead learn a general massaging policy that can quickly adjust to each client's needs based on proprioceptive and other feedback. However, the optimal approach may vary depending on the specific scenario, such as the number of clients and their variability in terms of therapeutic needs. This line of reasoning leads us to formulate a first hypothesis.

> **Hypothesis 1**
>
> When the reward and transition function share a structure across multiple tasks, task-agnostic approaches have the potential to outperform task-aware approaches in settings where task memorization is difficult, e.g., when the dimensionality and number of tasks increase, or when data and compute resources are limited.

## 3.3 BASELINES

**FineTuning** is a simple approach to a CL problem. It learns each incoming task without any mechanism to prevent forgetting. Its performance on past tasks indicates how much forgetting is incurred in a specific CL scenario.

**Experience Replay (ER)** accumulates data from previous tasks in a buffer for retraining purposes, thus slowing down forgetting (Rolnick et al., 2019; Aljundi et al., 2019; Chaudhry et al., 2019; Lesort, 2020). Although simple, it is often a worthy adversary for CL methods. One limitation of replay is that, to approximate the data distribution of all tasks, its compute requirements scale linearly with the number of tasks, leaving little compute for solving the current task, assuming a fixed compute budget. To achieve a better trade-off between remembering previous tasks and learning the current one, we use a strategy that caps replay by oversampling the data captured in the current task, as explained in Algorithm 1 L8-9. Note that to do this we maintain two separate buffers, which requires, a priori, being task-aware. However, the desired behavior can be achieved in a task-agnostic way by oversampling recently collected data.

**Multi-task (MTL)** trains on all tasks simultaneously and so it does not suffer from the challenges arising from learning on a non-stationary task distribution. It serves as a soft upper bound for CL methods.

**Independent** learns a set of separate models for each task whereby eliminating the CL challenges as well as the MTL ones, e.g., learning with conflicting gradients (Yu et al., 2020).

The aforementioned baselines are mixed and matched with the architectural choices (Section 3.2) to form different baselines, e.g. MTL with TaskID (MTL-TaskID) or FineTuning with MH (FineTuning-MH). At the core of this work lies a particular combination, explained next.

**Replay-based Recurrent RL (3RL)** A general approach to TACRL is to combine ER—one of CL's most versatile baselines—with an RNN, one of RL's most straightforward approaches to handling partial observability. We refer to this baseline as *replay-based recurrent RL* (3RL). As an episode unfolds, 3RL's RNN representations $z_t = \text{RNN}(\{(s_i, a_i, r_i)\}_{i=1}^{t-1})$ should predict the task with increasing accuracy, thus helping the actor $\pi_\theta(a|s,z)$ and critic $Q_\phi(s, a, z)$ in their respective approximations. We will see in Section 4, that the RNN delivers more than ex-

pected: it enables forward transfer by decomposing new tasks and placing them in the context of previous ones. We provide pseudocode for 3RL in Algorithm 1, which we kept agnostic to the base algorithm and not tied to episodic RL. as in Fakoor et al. (2020b); Ni et al. (2021), the actor and critics enjoy their own RNNs. They are thus parameterized by $\theta$ and $\phi$, respectively. Our RNN implementation employs gated recurrent units (GRUs) (Chung et al., 2014), as prescribed by Fakoor et al. (2020b).

Let us revisit the example of the massaging robot in a continual learning setting where the robot learns new clients incrementally. At a certain point, it may require more information bits to learn the specific nuances of a new client than to learn how to quickly adapt to handle them. This can lead to a situation where the task-aware approach results in more memory loss than the task-agnostic one. This reasoning leads to the second hypothesis.

---

**Algorithm 1:** 3RL in TACRL

**Environment:** a set of $K$ MPDs, allowed timesteps $T$
**Input:** initial parameters $\theta$, empty replay buffers $\mathcal{D}$ and $\mathcal{D}^{\text{old}}$ , replay cap $\beta$, batch size $b$, history length $h$

1 **for** *task $\tau$ in $K$* **do**
2      set environment to $\tau^{th}$ MDP
3      **for** *times-steps $t$ in $T$* **do**
         /* Sampling stage                                 */
4          compute dynamic task representation $z_t = \text{RNN}_\theta(\{(s_i^o, a_i, r_i)\}_{i=t-h-1}^{t-1})$
5          observe state $s_t^o$ and execute action $a_t \sim \pi_\theta(\cdot|s_t^o, z_t)$
6          observe reward $r_t$ and next state $s_{t+1}^o$
7          store $(s_t^o, a_t, r_t, s_{t+1}^o)$ in buffer $\mathcal{D}$
         /* Updating stage                                  */
8          sample a batch $B$ of $b \times min(\frac{1}{n}, 1 - \beta)$ trajectories from the current replay buffer $\mathcal{D}$
9          append to $B$ a batch of $b \times min(\frac{n-1}{n}, \beta)$ trajectories from the old buffer $\mathcal{D}^{\text{old}}$
10          Compute loss on $B$ and accordingly update parameters $\theta$ with one step of gradient descent
11      empty replay buffer $\mathcal{D}$ into $\mathcal{D}^{\text{old}}$

---

> **Hypothesis 2**
>
> In the context of continual learning, Hypothesis #1 may have even greater significance due to the reduced impact of catastrophic forgetting in algorithms that continually learn to adapt, in contrast to algorithms that attempt to memorize each task.

## 4    EMPIRICAL FINDINGS

We now present the empirical findings resulting from our task-agnostic continual RL (TACRL) experiments instantiated in both synthetic data and challenging robotic manipulation tasks.

**Benchmarks**    To evaluate our hypotheses, it is useful to have a mechanism for controlling the complexity of the available tasks. We propose a synthetic data benchmark that allows for the manipulation of the dimensions of observations and actions. The benchmark involves maximizing a multidimensional quadratic function, with each task having its own set of parameters for $A_\tau$, $b_\tau$, and $c_\tau$. The reward function for each task is defined as $r(s^o, \tau) = s^{o\top} A_\tau s^o + b_\tau s^o + c_\tau$, where $s^o$ is the observation vector and $\tau$ is the task. We sample $A_\tau$ to be negative definite to ensure a global maximum for the quadratic function, and we set $c_\tau$ such that all tasks have the same maximum reward. The transition function is fixed and defined as $s_{t+1}^o = s_t^0 + a_t$ where $a_t \in [-1, 1]$. Our benchmark provides a standardized platform for evaluating the performance of reinforcement learning algorithms in various task settings.

The second benchmark we study is Meta-World (Yu et al., 2019), the canonical evaluation protocol for multi-task reinforcement learning (MTRL) (Yu et al., 2020; Yang et al., 2020; Kumar et al., 2020; Sodhani et al., 2021). Meta-World offers a suite of 50 distinct robotic manipulation environments. What differentiates Meta-World from previous MTRL and meta-reinforcement learning benchmarks (Rakelly et al., 2019) is the broadness of its task distribution. Specifically, the different manipulation tasks are instantiated in the same state and action space[4] and share a reward structure, i.e., the reward functions are combinations of reaching, grasping, and pushing different objects with varying shapes, joints, and connectivity. Meta-World is thus fertile ground for algorithms to transfer skills across tasks while

---

[4]the fixed action space is an important distinction with traditional incremental supervised learning

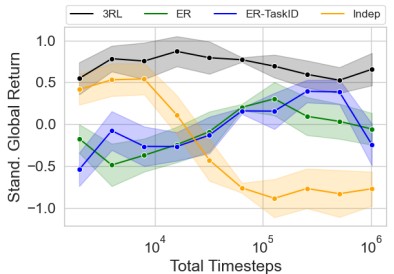 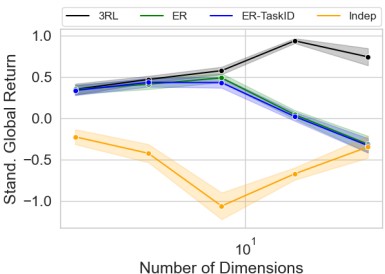 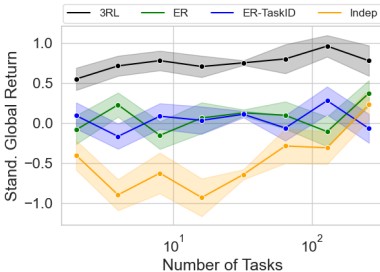

Figure 2: **Standardized Results (IQM) for the CL Synthetic Tasks (40K Runs)** For each combination of studied dimensions (total timesteps, number of dimensions, number of tasks), we have standardized the performance of the runs, resulting in standardized runs. Therefore, the plots reflect the ranking of the approach and not their absolute performance. We report the Interquartile mean at 50% of each method as a proxy for robustness. As expected, 3RL outperforms ER and TaskID in low timesteps regimes, ranks better as the number of observed dimensions increases. However, the superior performance of the RNN is independent of the number of tasks considered. The maximal performance plots and the plots with all methods are found in App. E.1 and App. E.2.

representing the types of tasks likely relevant for real-world RL applications (see Figure 1 for a rendering of some of the environments) and as a result its adoption in CRL is rapidly increasing (Wolczyk et al., 2021a; Mendez et al., 2020; Berseth et al., 2021). Benchmarking in this space is also an active research area (Mendez & Eaton, 2022).

We use a subset of Meta-World called `CW10`. It is a benchmark introduced in Wolczyk et al. (2021a) with a particular focus on *forward transfer*, namely, by comparing a method's ability to outperform one trained from scratch on new tasks. `CW10` is composed of a particular subset of Meta-World conducive to forward transfer and prescribes 1M steps per task, where a step corresponds to a sample collection and an update.

We also study a new benchmark composed of the 20 first tasks of Meta-World which we will use to explore a more challenging regime: the task sequence is twice as long and data and compute are constrained to half, i.e., 500k steps per task. We refer to this benchmark as `MW20`. We assembled all experiments with the Sequoia software for continual-learning research (Normandin et al., 2022).

For metrics, the global return/success and current return/success are the average success on all tasks and the average success on the task that the agent is currently learning, respectively.

**Experimental Details** To ensure controlled variation in scenario difficulty in our synthetic data experiments, we manipulate several parameters: the number of tasks, observation and action dimensions, and the total number of timesteps. To mitigate variations in performance across different scenarios, we standardize performance for each combination of these variables. Therefore, the performance plots we present should be interpreted relative to one another rather than as absolute measures of performance.

To optimize our results, we conducted an exhaustive random search of over 80,000 runs, varying hyperparameters for all possible methods. To report robustness, we calculate the interquantile mean (IQM) at 50%, meaning the top and bottom 25% of runs are excluded. For maximal performance, we report the mean of the top 10% of runs. The reported shaded areas are 2 standard errors. Full details of our experimental setup can be found in App. D.1.

As for the robotics tasks, we use the hyperparameters prescribed by Meta-World for their Multi-task SAC (MTL-SAC) method (see App. D.2). We ensured the performance of our SAC implementation on the `MT10`, one of Meta-World's prescribed MTRL benchmarks, matches theirs (see App. F). For an explanation as to why 1) our reported performances are lower than those from the original Continual World, and 2) our baselines struggle with tasks learned easily in the single-task learning (STL) regime from the original Meta-World paper (Yu et al., 2019), we refer to App. D.4. We test the methods using 8 seeds and report 90% T-test confidence intervals as the shaded area of the figures. As explained in algorithm 1, we oversample recently gathered data. When oversampling, we set the replay cap at 80%, thus always spending at least 20% of the compute budget on the current task.

## 4.1 CAN CONTINUAL FAST ADAPTATION OVERCOME CONTINUAL TASKS MEMORIZATION?

We present the initial results of our comprehensive synthetic empirical study in Figure 2. We compare a subset of continual learning approaches across a range of scenario complexities. A clear pattern emerges, with 3RL showing superior performance in terms of robustness. This is encouraging as 3RL has the potential to address the fragility

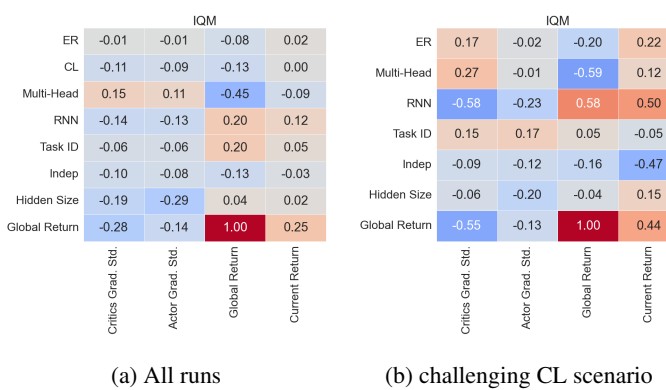

(a) All runs                    (b) challenging CL scenario

Figure 3: **Spearman Correlation Matrix of Synthetic Experiments (80k Runs)** The challenging CL scenario consists of 32 tasks, 32 dimensions, and 1M timesteps. Our analysis reveals that the RNN model enhances both robustness and maximal performance across studied slices, as indicated by the IQM (here) and top 10% (App. E.4) plots, respectively. The RNN model proves especially helpful in the challenging CL scenario, where it potentially achieves superior performance through a reduction in gradient conflict. Additionally, our findings show that the Task ID also improves robustness and maximal performance, although to a lesser extent and not through gradient conflict reduction.

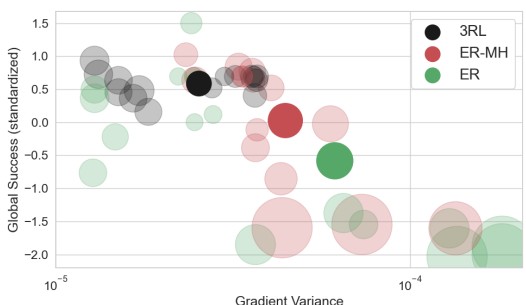

Figure 4: **In Meta-World, 3RL decreases gradient conflict leading to an increase in training stability and performance.** The global success and gradient as measured by the variance of the gradients shown are plotted against each other. Training instability as measured by the variance of the Q-values throughout learning is represented by the markers' size, in a log scale. Transparent markers depict seeds, whereas opaque ones depict the means. We observe a negative correlation between performance and gradient conflict (-0.75) as well as performance and training stability (-0.81), both significant under a 5% significance threshold. The hypothesis is that 3RL improves performance by reducing gradient conflict via dynamic task representations.

of deep reinforcement learning in real-world settings (Dulac-Arnold et al., 2019). Notably, 3RL outperforms other approaches in regimes with higher observation and action dimensionalities, as expected. This aligns with the intuition that as the dimensionality of a problem increases, it becomes more challenging to memorize all tasks (ER-TaskID) or find a single optimal solution to all tasks (ER), compared to learning a general solution that can adapt to specific tasks at test time (3RL).

In terms of maximal performance (see App. E.2), we do not observe any significant patterns in the results. It appears that for the synthetic benchmark, finding the appropriate hyperparameters is more critical than the choice of method to achieve maximal performance.

To gain further insights into the results and understand how 3RL outperforms the other approaches, we examine a challenging scenario in detail. This scenario consists of 32 tasks, in which we test the ability to learn in the highest number of observation and action dimensions studied (32) and to sustain a long training period (1 million timesteps). Such a setting aligns with real-world use cases as well as our experiments in Meta-world, thus representing an important test for the different approaches.

Furthermore, motivated by the well-known fact that optimizing multiple tasks simultaneously can often result in the emergence of conflicting gradients or task interference (Yu et al., 2020), we are compelled to investigate whether such phenomena could potentially impede robustness and maximal performance in the synthetic benchmark. Additionally, we conjecture that the RNN possesses an inherent capability to contextualize new tasks based on prior knowledge, thereby mitigating task interference and consequently exhibiting superior performance.

To test for this effect, we use the standard deviation of the gradients *averaged over the dimension of the parameters* on the mini-batch throughout the training as a proxy of task interference or gradient conflict. We explain in App. L why we use the gradients' standard deviation to measure gradient conflict instead of using the angle between the gradients as in Yu et al. (2020). We compute correlation matrices between different method attributes and metrics as shown in Figure 3. In the challenging continual learning scenario, we observe again that the RNN approach consistently outperforms the other approaches in terms of both robustness and maximal performance.

In line with our hypothesis, we observe that the RNN approach does indeed improves performance in the synthetic data benchmark by reducing gradient conflict, which has a negative correlation with performance. This finding sheds light on one mechanism by which the RNN approach enhances performance, especially in the challenging scenario. Additionally, our results suggest that gradient conflict may partially explain why the multi-head approach performs poorly in the synthetic data benchmark. More correlation matrices are found in App. E.4.

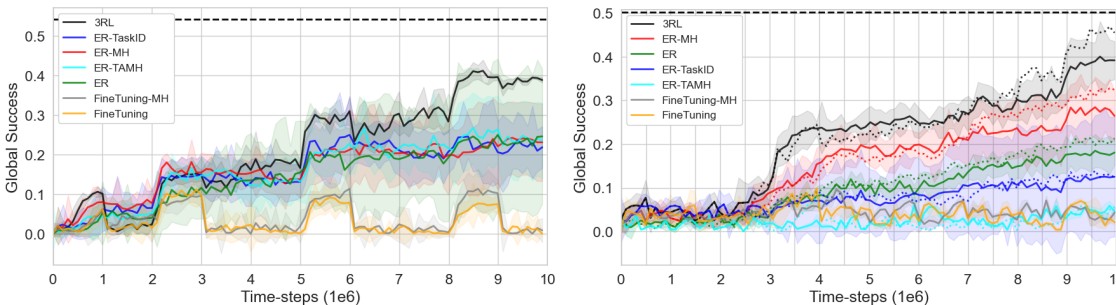

Figure 5: **3RL outperforms all baselines in both `CW10` (left) and `MW20` (right)**. The dashed horizontal black line at the top is the reference performance of training independent models on all tasks. The dotted lines (right plot) represent the methods' performance when they oversample recently collected data. 3RL outperforms all other task-agnostic and more interestingly task-aware baselines. As a side note, ER is high variance as it attempts to solve the POMDP directly without having any explicit or implicit mechanism to do so.

Our forthcoming experiments are performed in the domain of continual robotic manipulation tasks. We postulate that the well-known deadly triad issue (Sutton & Barto, 2018; Hasselt et al., 2018), which emerges from the combination of function approximation, bootstrapping, and off-policy learning, and potentially leads to divergence in value estimate, will prove to be a crucial problem in this context. The non-stationary task distribution, coupled with the intricate Meta-World environment, further exacerbates the issue. Consequently, we anticipate that 3RL, with its capacity to reduce gradient conflict, especially in lengthy and high-dimensional training regimes, may help improve stability and prove to be a superior CRL approach in more realistic tasks.

Our empirical evaluation compares 3RL to several baselines on the `CW10` and `MW20` benchmarks, as presented in Figure 5. In these results, 3RL outperforms all other competing methods on both robotic benchmarks. To study the deadly triad hypothesis, we conduct an analysis of global performance, gradient conflict, and training stability, which is presented in Figure 4. Our findings provide empirical support for the hypothesis that one mechanism through which the RNN improves performance is indeed reduced gradient conflict and increased stability in the Q values.

To strengthen the conclusion reached in the study, several additional observations specific to the robotic benchmarks are provided in the Appendix. One of the observations demonstrates that the difference in the number of parameters of the methods is not responsible for the results obtained (App. G). Another observation shows that combining task awareness with a recurrent neural network does not lead to improved performance (App. N). Furthermore, it is demonstrated that the RNN does not individually improve single-task performance (App. H). The mechanism responsible for the superior performance achieved by the proposed algorithm is not parameter stability (App. I), i.e. the tendency of a parameter to stay within its initial value while new knowledge is incorporated. Finally, some support is provided for the hypothesis that the RNN is able to place new tasks within the context of previous ones, thus facilitating forward transfer and improving optimization (App. J). This is backed up by qualitative evidence presented in Figure 1.

### 4.2 CONTRASTING CONTINUAL LEARNING AND MULTI-TASK LEARNING

We proceed to investigate our Hypothesis #2, which postulates that the advantages of fast adaptation over task memorization are magnified in a continual learning setting, due to the phenomenon of catastrophic forgetting. We conduct a comparative study between continual learning and multi-task learning methods and report our findings in Figure 6. As hypothesized, TaskID exhibits a significant decrease in performance with increasing dimensionality under the continual learning setting. Additionally, we observe that 3RL outperforms its multi-task learning counterpart in high-dimensional scenarios, which is a surprising finding since multi-task learning is often used as a soft upper bound in evaluating CL methods in both supervised (Aljundi et al., 2019; Lopez-Paz & Ranzato, 2017; Delange et al., 2021) and reinforcement learning (Rolnick et al., 2019; Traoré et al., 2019; Wolczyk et al., 2021a).

We end by comparing CRL and MTRL approaches in the robotic environment. In Figure 7 we report, for the second time, the results of the `MW20` experiments. This time, we focus on methods that oversample the current task and more importantly, we report the performance of each method's multi-task analog, i.e. their soft-upper bound (dotted line of the same color). Note that FineTuning methods do not have an MTRL counterpart and are thus not included in the current analysis. 3RL is the only approach that matches the performance of its MTRL equivalent. We believe it is the first time that a specific method achieves the same performance in a non-stationary task regime compared to the stationary one, amidst the introduced challenges like forgetting.

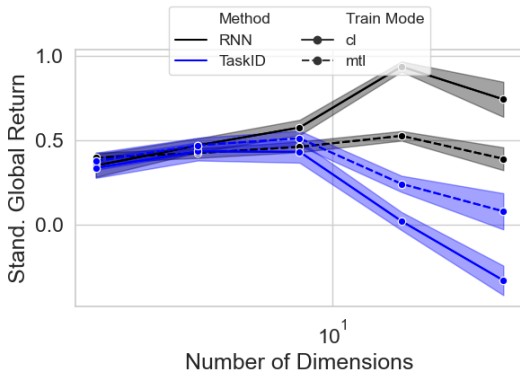

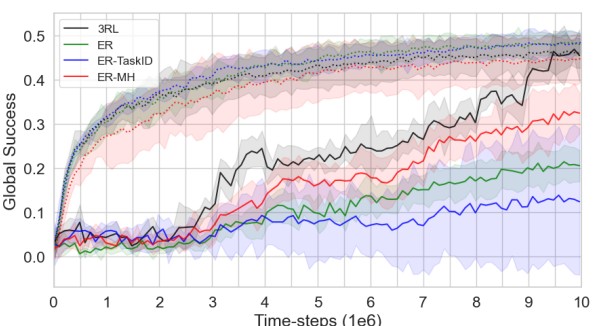

Figure 6: **Standardized Results (IQM) for the CRL and MTRL Synthetic Tasks (80k Runs)**. Aligned with hypothesis #2, 3RL's outperformance over tasks memorization is amplified in in continual learning as the complexity of the tasks increases and might be caused by fast adaptation.

Figure 7: **3RL reaches its MTRL soft-upper bound**. Continual vs multi-task learning methods in solid lines and dotted lines, respectively. 3RL methods match its soft-upper bound MTL analog as well as the other MTRL baselines. In contrast, other the performance of other baselines is drastically hindered by the non-stationary task distribution.

## 5 RELATED WORK

To study CRL in realistic settings, Wolczyk et al. (2021b) introduce the Continual World benchmark and discover that many CRL methods that reduce forgetting lose their transfer capabilities in the process, i.e. that policies learned from scratch generally learn new tasks faster than continual learners. Previous works study CL and compare task-agnostic methods to their upper bounds (Zeno et al., 2019; van de Ven & Tolias, 2019) as well as CL methods compare to their multi-task upper bound (Ribeiro et al., 2019; Rolnick et al., 2019; Ammar et al., 2014). Refer to Khetarpal et al. (2020) for an in-depth review of continual RL as well as Lesort et al. (2021); Hadsell et al. (2020) for a CL in general.

Closer to our training regimes, TACRL is an actively studied field. Xu et al. (2020) uses an infinite mixture of Gaussian Processes to learn a task-agnostic policy. Kessler et al. (2021) learns multiple policies and casts policy retrieval as a multi-arm bandit problem. As for Berseth et al. (2021); Nagabandi et al. (2019), they use meta-learning to tackle the task-agnosticism part of the problem.

RNNs were used in the context of continual supervised learning in the context of language modeling (Wolf et al., 2018; Wu et al., 2021) as well as in audio (Ehret et al., 2020; Wang et al., 2019). We refer to Cossu et al. (2021) for an in-depth review of RNN in continual supervised learning.

As in our work, RNN models have been effectively used as policy networks for reinforcement learning, especially in POMDPs where they can effectively aggregate information about states encountered over time (Wierstra et al., 2007; Fakoor et al., 2020b; Ni et al., 2021; Heess et al., 2015). RNNs were used in the context of MTRL Nguyen & Obafemi-Ajayi (2019). Closer to our work, Sorokin & Burtsev (2019) leverages RNNs in a task-aware way to tackle a continual RL problem. To the best of our knowledge, RNNs have not been employed within TACRL nor combined with Experience Replay in the context of CRL.

## 6 CONCLUSION

Our study demonstrates that incorporating a recurrent memory into task-agnostic continual reinforcement learning enables TACRL methods, such as 3RL, to match or even surpass their multi-task upper bound while outperforming similar task-aware methods. Our extensive experiments support the hypothesis that 3RL is the preferred CRL approach when data and computation are limited or when the problem's dimensionality increases. A possible explanation for the improved performance is the reduction in gradient conflict, which stabilizes training in terms of Q-value variance.

Our findings challenge the conventional assumption that TACRL is inherently more challenging than task-aware multi-task RL. Despite being more broadly applicable, TACRL methods like 3RL can perform just as well as their task-aware and multi-task counterparts. While further research is needed to reach a definitive conclusion, the need for CL-developed forgetting-alleviating and task-inference tools for CRL is now being questioned.

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

# Appendix: Task-Agnostic Continual Reinforcement Learning: Gaining Insights and Overcoming Challenges

## A    EXTENDING TACRL'S RELATED SETTINGS

We continue the discussion on TACRL's related settings. In Meta-RL (Finn et al., 2017), the training task, or analogously the training hidden-states $s^h$, are different from the testing ones. We thus separate them into disjoint variable $s_{\text{train}}^h$ and $s_{\text{test}}^h$. In this setting, some fast adaptation to $s^h$ is always required. Meta-RL does not deal with a non-stationary training task distribution. Its continual counterpart however, i.e. Continual Meta-RL (Berseth et al., 2021), does. Table 2 summarizes the settings.

Noteworthy, the Hidden Parameter MDP (HiP-MDP) (Doshi-Velez & Konidaris, 2016) is similar to the HM-MDP but assumes a hidden states, i.e., the hidden states are samples i.i.d. Another setting similar to the HM-MD is the Dynamic Parameter MDP (Xie et al., 2020) in which the hidden-state are non-stationary but change at every episode.

| | T | $\pi$ | Objective | Evaluation |
|---|---|---|---|---|
| MDP (Sutton & Barto, 2018) | $p(s_{t+1}|s_t, a_t)$ | $\pi(a_t|s_t)$ | $\mathbb{E}_\pi\left[\sum_{t=0}^\infty \gamma^t r_t\right]$ | - |
| POMDP (Kaelbling et al., 1998) | $p(s_{t+1}^h, s_{t+1}^o|s_t^h, s_t^o, a_t)$ | $\pi(a_t|s_{1:t}^o, a_{1:t-1}, r_{1:t-1})$ | $\mathbb{E}_{s^h}\left[\mathbb{E}_\pi\left[\sum_{t=0}^\infty \gamma^t r_t\right]|s^h\right]$ | - |
| HM-MDP (Choi et al., 2000) | $p(s_{t+1}^o|s_{t+1}^h, s_t^o, a_t)p(s_{t+1}^h|s_t^h)$ | $\pi(a_t|s_{1:t}^o, a_{1:t-1}, r_{1:t-1})$ | $\mathbb{E}_{s^h}\left[\mathbb{E}_\pi\left[\sum_{t=0}^\infty \gamma^t r_t\right]|s^h\right]$ | - |
| **Task-agnostic CRL** | $p(s_{t+1}^o|s_{t+1}^h, s_t^o, a_t)p(s_{t+1}^h|s_t^h)$ | $\pi(a_t|s_{1:t}^o, a_{1:t-1}, r_{1:t-1})$ | $\mathbb{E}_{s^h}\left[\mathbb{E}_\pi\left[\sum_{t=0}^\infty \gamma^t r_t\right]|s^h\right]$ | $\mathbb{E}_{\tilde{s}^h}\left[\mathbb{E}_\pi\left[\sum_{t=0}^\infty \gamma^t r_t\right]|s^h\right]$ |
| Task-Aware CRL | $p(s_{t+1}^o|s_{t+1}^h, s_t^o, a_t)p(s_{t+1}^h|s_t^h)$ | $\pi(a_t|s_t^h, s_t^o)$ | $\mathbb{E}_{s^h}\left[\mathbb{E}_\pi\left[\sum_{t=0}^\infty \gamma^t r_t\right]|s^h\right]$ | $\mathbb{E}_{\tilde{s}^h}\left[\mathbb{E}_\pi\left[\sum_{t=0}^\infty \gamma^t r_t\right]|s^h\right]$ |
| Multi-task RL | $p(s_{t+1}^o|s_{t+1}^h, s_t^o, a_t)p(s_{t+1}^h)$ | $\pi(a_t|s_t^h, s_t^o)$ | $\mathbb{E}_{\tilde{s}^h}\left[\mathbb{E}_\pi\left[\sum_{t=0}^\infty \gamma^t r_t\right]|s^h\right]$ | - |
| Meta RL (Finn et al., 2017) | $p(s_{t+1}^o|s_{t+1}^h, s_t^o, a_t)p(s_{t+1}^h)$ | $\pi(a_t|s_{1:t}^o, a_{1:t-1}, r_{1:t-1})$ | $\mathbb{E}_{s_{\text{train}}^h}\left[\mathbb{E}_\pi\left[\sum_{t=0}^\infty \gamma^t r_t\right]|s_{\text{train}}^h\right]$ | $\mathbb{E}_{\tilde{s}_{\text{test}}^h}\left[\mathbb{E}_\pi\left[\sum_{t=0}^\infty \gamma^t r_t\right]|s_{\text{test}}^h\right]$ |
| Continual Meta-RL | $p(s_{t+1}^o|s_{t+1}^h, s_t^o, a_t)p(s_{t+1}^h|s_t^h)$ | $\pi(a_t|s_{1:t}^o, a_{1:t-1}, r_{1:t-1})$ | $\mathbb{E}_{s_{\text{train}}^h}\left[\mathbb{E}_\pi\left[\sum_{t=0}^\infty \gamma^t r_t\right]|s_{\text{train}}^h\right]$ | $\mathbb{E}_{\tilde{s}_{\text{test}}^h}\left[\mathbb{E}_\pi\left[\sum_{t=0}^\infty \gamma^t r_t\right]|s_{\text{test}}^h\right]$ |

Table 2: Summarizing table of the settings relevant to TACRL. For readability purposes, $\tilde{s}^h$ denotes the stationary distribution of $s^h$. The Evaluation column if left blank when it is equivalent to the Objective one.

## B    SOFT-ACTOR CRITIC

The Soft Actor-Critic (SAC) (Haarnoja et al., 2018b) is an off-policy actor-critic algorithm for continuous actions. SAC adopts a maximum entropy framework that learns a stochastic policy which not only maximizes the expected return but also encourages the policy to contain some randomness. To accomplish this, SAC utilizes an actor/policy network (i.e. $\pi_\phi$) and critic/Q network (i.e. $Q_\theta$), parameterized by $\phi$ and $\theta$ respectively. Q-values are learnt by minimizing one-step temporal difference (TD) error by sampling previously collected data from the replay buffer (Lin, 1992) denoted by $\mathcal{D}$.

$$\mathcal{J}_Q(\theta) = \mathbb{E}_{s,a}\left[\left(Q_\theta(s,a) - y(s,a)\right)^2\right], \; a' \sim \pi_\phi(\cdot|s') \tag{1}$$

where $y(s,a)$ is defined as follows:

$$y(s,a) = r(s,a) + \gamma \mathbb{E}_{s',a'}\left[Q_{\hat{\theta}}(s',a') - \alpha \log(a'|s')\right]$$

And then, the policy is updated by maximizing the likelihood of actions with higher Q-values:

$$\mathcal{J}_\pi(\phi) = \mathbb{E}_{s,\hat{a}}\left[Q_\theta(s,\hat{a}) - \alpha \log \pi_\phi(\hat{a}|s)\right], \; \hat{a} \sim \pi_\phi(\cdot|s) \tag{2}$$

where $(s,a,s') \sim \mathcal{D}$ (in both equation 1 and equation 2) and $\alpha$ is entropy coefficient. Note that although SAC is used in this paper, other off-policy methods for continuous control can be equally utilized for CRL. SAC is selected here as it has a straightforward implementation and few hyper-parameters.

## C  BASELINES DEFINITIONS

**FineTuning-MH** is FineTuning with task-specific heads. For each new task, it spawns and attaches an additional output head to the actor and critics. Since each head is trained on a single task, this baseline allows to decompose forgetting happening in the representation of the model (trunk) compared to forgetting in the last prediction layer (head). It is a task-aware method.

**ER-TaskID** is a variant of ER that is provided with task labels as inputs (i.e. each observation also contains a task label). It is a task-aware method that has the ability to learn a task representation in the first layer(s) of the model.

**ER-MH** is ER strategy which spawns tasks-specific heads (Wolczyk et al., 2021a), similar to FineTuning-MH. ER-MH is often the hardest to beat task-aware baseline (). ER-TaskID and ER-MH use two different strategies for modelling task labels. Whereas, MH uses $|h| \times |A|$ task-specific parameters (head) taskID only uses $|h|$, with $|h|$ the size of the network's hidden space (assuming the hidden spaces at each layer have the same size) and $|A|$ the number of actions available to the agent.

**ER-TAMH (task-agnostic multi-head)** is similar to ER-MH, but the task-specific prediction heads are chosen in a task-agnostic way. Specifically, the number of heads is fixed a priori (in the experiments we fix it to the number of total tasks) and the most confident actor head, w.r.t. the entropy of the policy, and most optimistic critic head are chosen. ER-TAMH has the potential to outperform ER, another task-agnostic baseline, if it can correctly infer the tasks from the observations.

**MTL** is our backbone algorithm, namely SAC, trained via multi-task learning. It is the analog of ER.

**MTL-TaskID** is MTL, but the task label is provided to the actor and critic. It is the analog of ER-TaskID and is a standard method, e.g. (Haarnoja et al., 2018a).

**MTL-MH** is MTL with a task-specific prediction network. It is the analog of ER-MH and is also standard, e.g. (Yu et al., 2020; Haarnoja et al., 2018a; Yu et al., 2019).

**MTL-TAMH** is similar to MTL-MH, but the task-specific prediction heads are chosen in the same way as in ER-TAMH.

**MTL-RNN** is similar to MTL, but the actor and critic are mounted with an RNN. It is the analog of 3RL.

## D  EXPERIMENTAL DETAILS

### D.1  SYNTHETIC DATA HYPERPARAMETERS

In Table 3 we report the hyperparameters ranges used for the synthetic benchmark experiments.

### D.2  META-WORLD HYPERPARAMETERS AND THEIR JUSTIFICATION

The choice of hyperparameters required quite some work. Initially, we used the SAC hyperparameters prescribed by Continual World (Wolczyk et al., 2021a), designed on Meta-World v1, without any success. We then tried some of Meta-World v2's prescribed hyperparameter, which helped us match MetaWorld's multi-task reported results.

However, the continual and multi-task learning baselines would still suffer from largely unstable training due to the deadly triad (van Hasselt et al., 2018) problems in CRL and MTRL. After further experimentation, we observed that gradient clipping could stabilize training, and that clipping the gradients to a norm of 1 achieved the desired behavior across all methods, except for the multi-head baseline in which 10 was more appropriate.

Lastly, we use automatic entropy tuning except in the MTRL experiments, where we found its omission to be detrimental. Because their MT-SAC implementation learns a task-specific entropy term, we think this is the reason why Yu et al. (2019) do not observe the same behavior. All hyperparameters are summarized in Table 4.

### D.3  COMPUTING RESOURCES

For the synthetic benchmark, we ran 82,160 different runs. We used 2 or 4 CPU per runs, resulting in 10,260 days worth of compute.

For the Meta-world benchmarks, all experiments were performed on Amazon EC2's P2 instances which incorporates up to 16 NVIDIA Tesla K80 Accelerators and is equipped with Intel Xeon 2.30GHz cpu family. All meta-world

| Setting | |
|---|---|
| total timesteps | [2,000, 4,000, 8,000, 16,000, 32,000, 64,000, 128,000, 256,000, 512,000, 1,024,000] |
| number of tasks | [2, 4, 8, 16, 32, 64, 128, 256] |
| number of dimensions | [2, 4, 8, 16, 32] |
| CL or MTL | [CL, MTL] |
| episode length | 100 |
| | |
| **All Methods** | |
| Architecture | 2-layer MLP |
| activation | ReLU |
| soft-target interpolation | $5 \times 10^{-3}$ |
| learning rate | log-uniform(min=0.00001, max=0.1) |
| batch size | uniform(min=2, max=256) |
| warm-up period | uniform(min=2, max=100) |
| burn-in period | uniform(min=2, max=100) |
| hidden size | [8, 16, 32, 64, 128] |
| automatic entropy tuning | [on, off] |
| | |
| **ER methods** | |
| replay cap $\beta$ | 1.0, 0.8, 0.5 |
| buffer size | [1,000, 10,000, 100,000, 1,000,000] |
| | |
| **methods with RNN** | |
| context size | [30, 50] |
| history length | [2, 4, 8, 16] |

Table 3: **Table of hyperparameters for the synthetic benchmark hyperparameter search.**

| Architecture | 2-layer MLP | | |
|---|---|---|---|
| hidden state | [400, 400] | | |
| activation | ReLU | | |
| episode length | Continual World: 200, else: 500 | | |
| minimum buffer size | 10 tasks: 1500, 20 tasks: 7500 | | |
| batch size | | | |
| learning rate | $1 \times 10^{-3}$ | | |
| soft-target interpolation | $5 \times 10^{-3}$ | | |
| RNN's context length | 15 | | |
| burn in period | 10,000 steps | | |
| | **Independent** | **MTRL** | **CRL** |
| automatic-entropy tuning | on | off | on |
| gradient clipping | None | MH: 10, else: 1 | MH: 10, else: 1 |

Table 4: **Table of hyperparameters.** The top hyperparameters are global, whereas the bottom ones are setting specific.

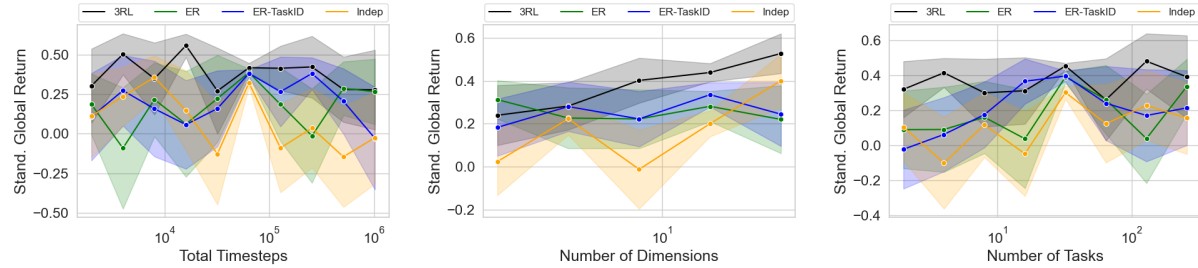

Figure 8: **standardized results (top 10% of the runs) for the CL synthetic tasks (40k runs)** Unlike for robustness, the impact of hyperpameter choice seems to overshadow the methods' impact on maximal performance.

experiments included in the paper can be reproduced by running 43 method/setting configurations with 8 seeds, each running for 4.2 days on average.

### D.4 RESOLVING THE CONFUSION ABOUT WHAT COULD SEEM AS A RESULT MISMATCH WITH OTHER META-WORLD EXPERIMENTS IN THE LITERATURE

The astute reader will find that: 1) our reported performances are lower than those from the original Continual World, and 2) our baselines struggle with tasks learned easily in the single-task learning (STL) regime from the original Meta-World paper Yu et al. (2019). These discrepancies are explained by the fact that Wolczyk et al. (2021a) carried out their original Continual-World study using Meta-World v1, which is far from the updated v2 version we use here. For example, the state space is thrice larger in v2 and the reward functions have been completely rewritten. The original Meta-World study of Yu et al. (2019) was performed under far more generous data and compute settings (see App. F) and relied on MTL and STL specific architectures and hyperparameters.

### D.5 SOFTWARE AND LIBRARIES

In the codebase we've used to run the experiments, we have leverage some important libraries and software. We used Mujoco (Todorov et al., 2012) and Meta-World (Yu et al., 2019) to run the benchmarks. We used Sequoia (Normandin et al., 2022) to assemble the particular CRL benchmarks, including `CW10`. We used Pytorch (Paszke et al., 2019) to design the neural networks.

## E SYNTHETIC DATA EXPERIMENTS

### E.1 TOP 10% OF RUNS IN THE CL SYNTHETIC DATA BENCHMARK FOR THE METHODS PRESENTED IN THE MAIN SECTION

See figure 8 for our proxy of maximal performance in the CL synthetic benchmark, i.e. the average of the top 10% runs.

### E.2 COMPLETE RESULTS FOR THE CL SYNTHETIC DATA EXPERIMENTS

See figure 9 for the complete CL synthetic data experiments, i.e. when all methods are included.

### E.3 COMPLETE RESULTS CONTRASTING CL AND MTL IN THE SYNTHETIC DATA EXPERIMENTS

See figure 10 for the complete results contrasting CL and MTL in the synthetic data experiments.

### E.4 MORE CORRELATION MATRICES IN THE SYNTHETIC EXPERIMENTS

See figure 11 for more correlation matrices in the synthetic experiment.

## F    VALIDATING OUR SAC IMPLEMENTATION ON MT10

In Figure 12 we validate our SAC implementation on Meta-World v2's MT10.

## G    OBSERVATION IN META-WORLD #1: LARGER NETWORKS DO NOT IMPROVE PERFORMANCE IN META-WORLD

In figure 13 we report that increasing the neural net capacity does not increase performance.

## H    OBSERVATIONS IN META-WORLD #2: RNN DOESN'T INDIVIDUALLY IMPROVES THE SINGLE-TASK PERFORMANCE

A simple explanation is that the RNN enhances SAC's ability to learn each task independently, perhaps providing a different inductive bias beneficial to each individual task. To test this hypothesis, we run the `CW10` benchmark this time with each task learned separately, which we refer to as the Independent baselines. Note that this hypothesis is unlikely since the agent observes the complete state which is enough to act optimally (i.e. the environments are MDPs not POMDPs). Unsurprisingly, we find the RNN decreases the performance of standard SAC by 8.2% on average on all tasks. Accordingly, we discard this hypothesis. figure 14 provides the complete STL results.

## I    OBSERVATION IN META-WORLD #2: RNN DOESN'T INCREASES PARAMETER STABILITY

The plasticity-stability tradeoff is at the heart of continual learning: plasticity eases the learning of new tasks. Naive learning methods assume stationary data and so are too plastic in non-stationary regimes leading to catastrophic forgetting. To increase stability, multiple methods enforce (Mallya & Lazebnik, 2018) or regularize for (Kirkpatrick et al., 2017b) *parameter stability*, i.e., the tendency of a parameter to stay within its initial value while new knowledge is incorporated. Carefully tuned task-aware methods, e.g. PackNet (Mallya & Lazebnik, 2018) in Wolczyk et al. (2021a), have the ability to prevent forgetting.[5]

Considering the above, we ask: could 3RL implicitly increase parameter stability? To test this hypothesis we measure the average total movement of the each weight throughout an epoch of learning which is defined by all updates in between an episode collection. To produce a metric suitable for comparing different runs which could operate in different regimes of parameter updates, we suppose report an entropy-like metric computed as follow $H_e = -(|\theta_{e+1} - \theta_e|)^\intercal \log(|\theta_{e+1} - \theta_e|)$, where $e$ is the epoch index. Note that the proposed metric takes all its sense when used relatively, (i.e. to compare the relative performance in-between methods) and not as an absolute measure of parameter stability Details about this experiment are found in App. K.

Figure 15 reports our proxy of total parameter movement for 3RL, ER, and ER-MH. We use these baselines since the gap between ER and its upper bound is the largest and the ER-MH gap is in between ER's and 3RL's. We find strong evidence to reject our hypothesis. After an initial increase in parameter stability, weight movement increases as training proceeds across all methods and even spikes when a new task is introduced (every 500K steps). MTL-RNN follows the same general pattern as the ER methods. Note that, it is not impossible that the *function* represented by the neural networks are stable even though their parameters are not. Nevertheless, it would be surprising that such a behaviour arose only in 3RL given the high similarity of the stability reported across all methods.

## J    OBSERVATION IN META-WORLD #3: RNN CORRECTLY PLACES THE NEW TASKS IN THE CONTEXT OF PREVIOUS ONES, ENABLING FORWARD TRANSFER AND IMPROVING OPTIMIZATION

As in real robotic use-cases, MW tasks share a set of low-level reward components like grasping, pushing, placing, and reaching, as well as set of object with varying joints, shapes, and connectivity. As the agent experiences a new task, the RNN could quickly infer how the new data distribution relates with the previous ones and provide to the actor and critics a useful representation. Assume the following toy example: task one's goal is to grasp a door handle, and task two's to open a door. The RNN could infer from the state-action-reward trajectory that the second task is composed

---

[5]The observation that PackNet outperforms *an* MTRL baseline in Wolczyk et al. (2021a) is different from our stronger observation that a *single* method, namely 3RL, achieves the same performance in CRL than in MTRL

of two subtasks: the first one as well as a novel pulling one. Doing so would increase the policy learning's speed, or analogously enable forward transfer.

Now consider a third task in which the agent has to close a door. Again, the first part of the task consists in grasping the door handle. However, now the agent needs to subsequently push and not pull, as was required in task two. In this situation, task interference Yu et al. (2020) would occur. Once more, if the RNN could dynamically infer from the context when pushing or pulling is required, it could modulate the actor and critics to have different behaviors in each tasks thus reducing the interference. Note that a similar task interference reduction should be achieved by task-aware methods. E.g., a multi-head component can enable a method to take different actions in similar states depending on the tasks, thus reducing the task interference.

Observing and quantifying that 3RL learns a representation space in which the new tasks are correctly *decomposed* and placed within the previous ones is challenging. Our initial strategy is to look for effects that should arise if this hypothesis was true (so observing the effect would confirm the hypothesis).

First, we take a look at the time required to adapt to new tasks: if 3RL correctly infers how new tasks relate to previous ones, it might be able to learn faster by re-purposing learned behaviors. Figure 16 depicts the current performance of different methods throughout the learning of CW10. For reference, we provide the results of training separate models, which we refer to as Independent and Independent RNN.

The challenges of Meta-World v2 compounded with the ones from learning multiple policies in a shared network, and handling an extra level of non-stationary, i.e. in the task distribution, leaves the continual learners only learning task 0, 2, 5, and 8. On those tasks (except the first one in which no forward transfer can be achieved) 3RL is the fastest continual learner. Interestingly, 3RL showcases some forward transfer by learning faster than the Independent methods on those tasks. This outperformance is more impressive when we remember that 3RL is spending most of its compute replaying old tasks. We thus find some support for Hypothesis #3.

Second, simultaneously optimizing for multiple tasks can lead to conflicting gradients or task interference (Yu et al., 2020). To test for this effect, we use the average variance of the gradients on the mini-batch throughout the training as a proxy of gradient conflict. We explain in L why we use the gradients' variance to measure gradient conflict instead of using the angle between the gradients as in Yu et al. (2020).

In figure 4 we show the normalized global success metric plotted against the gradient variance. In line with our intuition, we do find that the RNN increases gradient agreement over baselines. As expected, adding a multi-head scheme can also help, to a lesser extent. We find a significant negative correlation of -0.75 between performance and gradient conflict. App. M reports the evolution of gradient conflict through time in the actor and critic networks.

Figure 4 also reports training stability, as measured by the standard deviation of Q-values throughout training (not to be confused with the parameter stability, at the center of Hypothesis #2, which measures how much the parameters move around during training). We find 3RL enjoys more stable training as well as a the significant negative correlation of -0.81 between performance and training stability. Note that The plausibility of Hypothesis #3 is thus further increased.

We wrap up the hypothesis with some qualitative support for it. Figure 1 showcases the RNN representations as training unfolds. If the RNN was merely performing task inference, we would observe the trajectories getting further from each other, not intersecting, and collapsing once the task is inferred. Contrarily, the different task trajectories constantly evolve and seem to intersect in particular ways. Although only qualitative, this observation supports the current hypothesis.

## K  GRADIENT ENTROPY EXPERIMENT

To assess parameter stability, we look at the entropy of the parameters for each epoch. Because the episodes are of size 500, the epoch corresponds to 500 updates. To remove the effect of ADAM (Kingma & Ba, 2017), our optimizer, we approximate the parameters' movement by summing up their absolute gradients throughout the epoch. To approximate the sparsity of the updates, we report the entropy of the absolute gradient sum. For example, a maximum entropy would indicate all parameters are moving equally. If the entropy drops, it means the algorithm is applying sparser updates to the model, similarly to PackNet.

## L  GRADIENT VARIANCE AS A PROXY FOR GRADIENT CONFLICT

Yu et al. (2020) instead measure the conflict between two tasks via the angle between their gradients, proposing that the tasks conflict if this angle is obtuse. However we argue that it also critical to consider the magnitude of the gradients.

Consider an example of two tasks involving 1D optimization with two possible cases. In case 1, suppose that the gradient of the first task is 0.01 and the gradient of the second is -0.01. In case 2, suppose that the gradients are now 0.01 and 0.5, respectively. Measuring conflict via the angle would lead us to think that the tasks are in conflict in case 1 and are not in case 2. However, their agreement is actually much higher in case 1: both tasks agree that they should not move too far from the current parameter. In case 2, although the two tasks agree on the direction of the parameter update step, they have loss landscapes with great differences in curvature. The update step will be too small or too large for one of the tasks. We thus find variance to be a better measure of gradient conflict.

## M  GRADIENT CONFLICT THROUGH TIME

See figure 17.

## N  TASK-AWARE MEETS TASK-AGNOSTIC

In figure 18 we show that combining the RNN with MH is not a good proposition in `CW10`.

## O  LIMITATIONS

In this section we surface some limitations in our work. First of all, given the highly-demanding computational nature of the studied benchmarks, we could not run extensive hyperparameter searches as well as all the desired ablations. We have mostly relied on the hyperparameters prescribed by Meta-World, with the exception of the introduction of gradient clipping (App. D) which we found detrimental for the continual and multi-task learners to perform adequately. Importantly, we have not tested multiple context length for 3RL, an important hyperparameter for model-free recurrent RL (Ni et al., 2021). We can thus hypothesize that our 3RL's performances are underestimation.

Second, although Meta-World is know to be challenging, one could argue that its task-inference component is uncomplicated. Future work could explore task-agnostic continual RL benchmarks in which the different MDP are less distinctive.

Third, the hypothesis about the RNN correctly placing the new tasks in the context of previous ones (Hypothesis #3) is difficult to test. We have tested for effects that should arise if the hypothesis is true, but those effect could arise for different reasons. Future work could potentially dismantle the reward functions and look for RNN context overlaps across tasks when a particular reward component, e.g., grasping a door knob, is activated.

Lastly, we have only investigated a single benchmark. Introducing more benchmarks would was outside our compute budget. Moreover, challenging RL benchmarks have steep user learning curve, especially in continual RL. We leave for future work the study of 3RL in a wide suite of benchmarks.

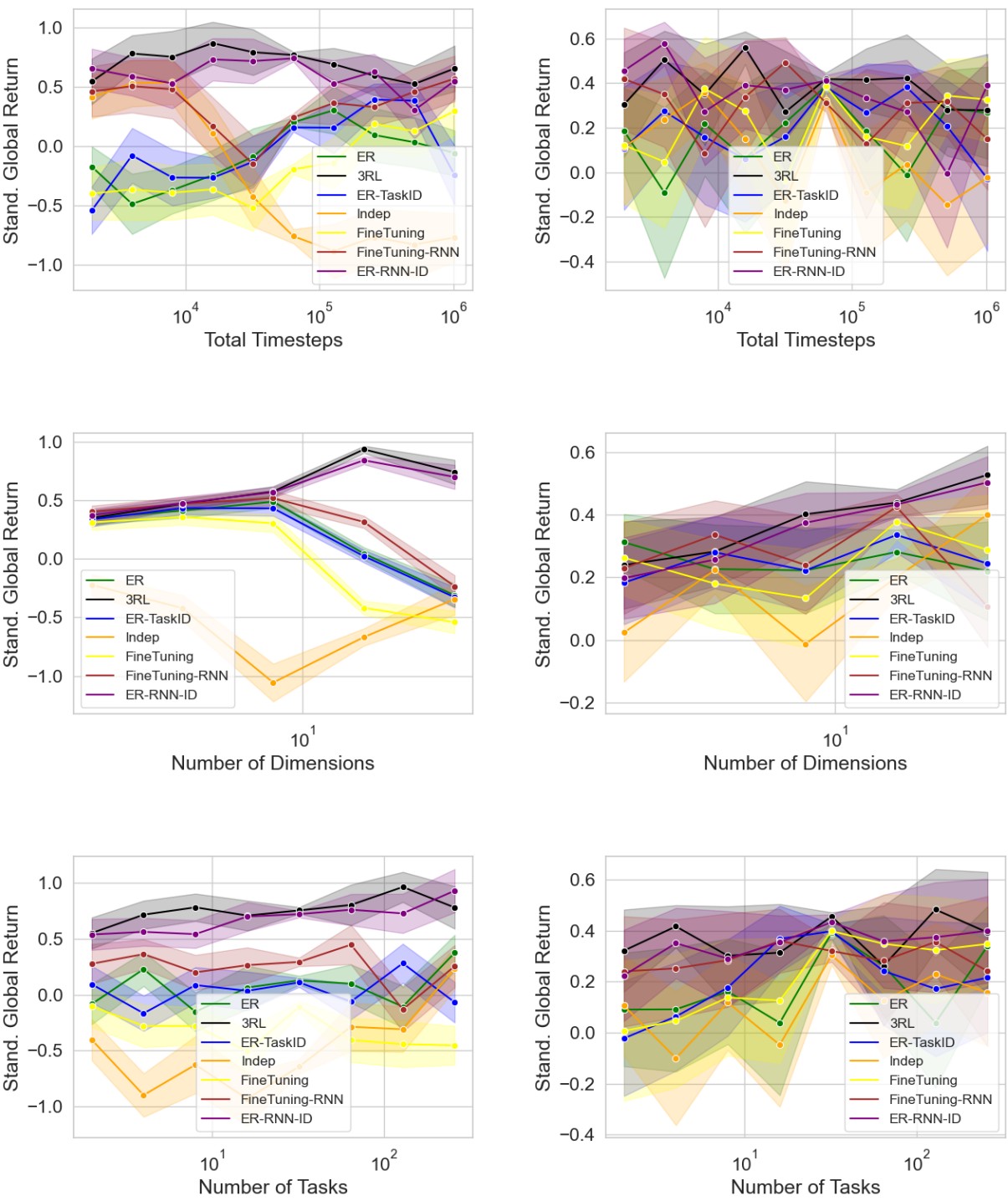

Figure 9: **Complete results for the CL synthetic experiments (40k runs)** For readability, we removed the multi-head (MH) methods from the plots, as they are performing relatively too low to others. See figure 10 for some MH results. On the left are IQM plots whereas on the right top 10% ones.

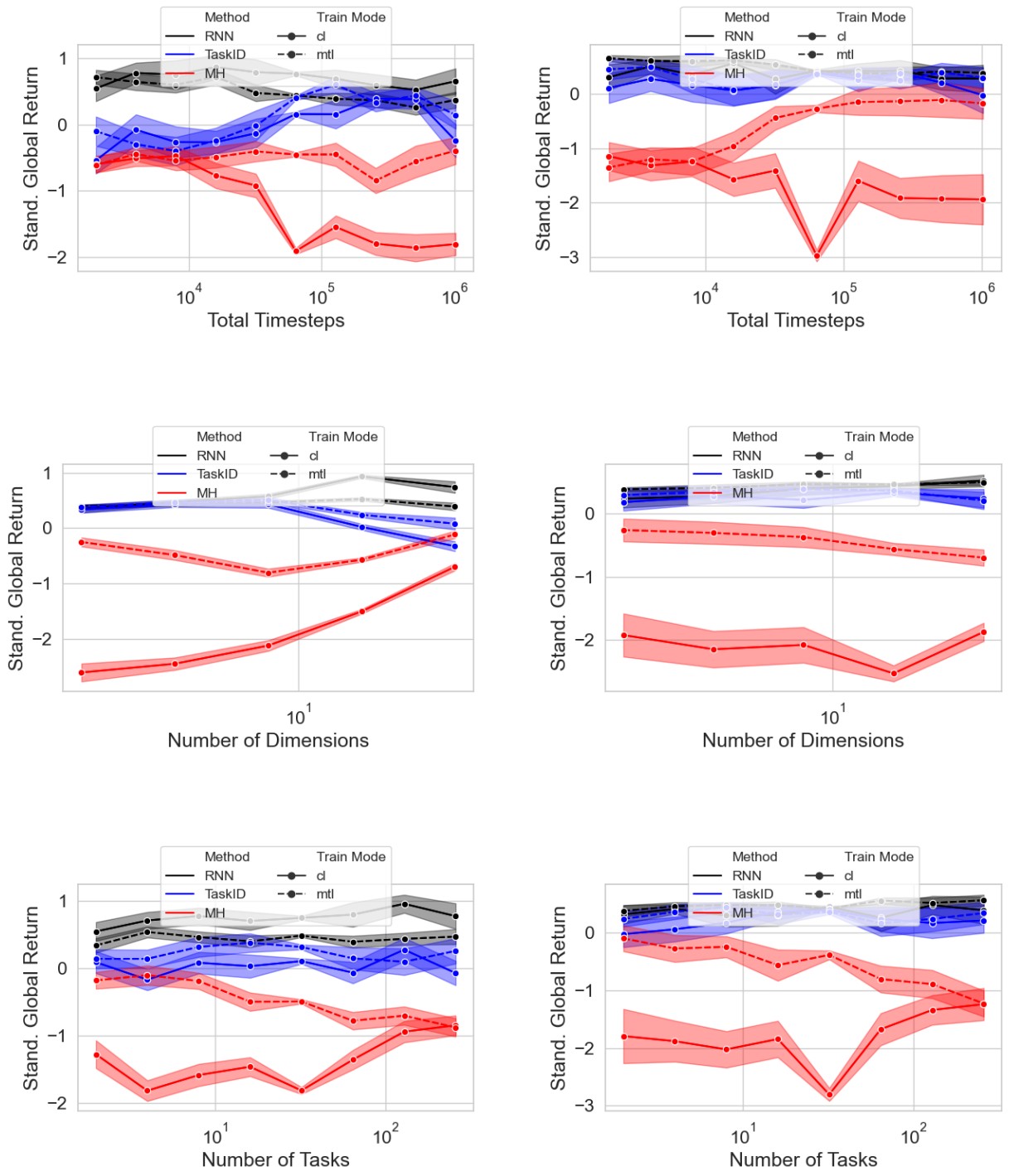

Figure 10: **Complete results for the CL vs MTL synthetic experiments (80k runs).** On the left are IQM plots whereas on the right top 10% ones.

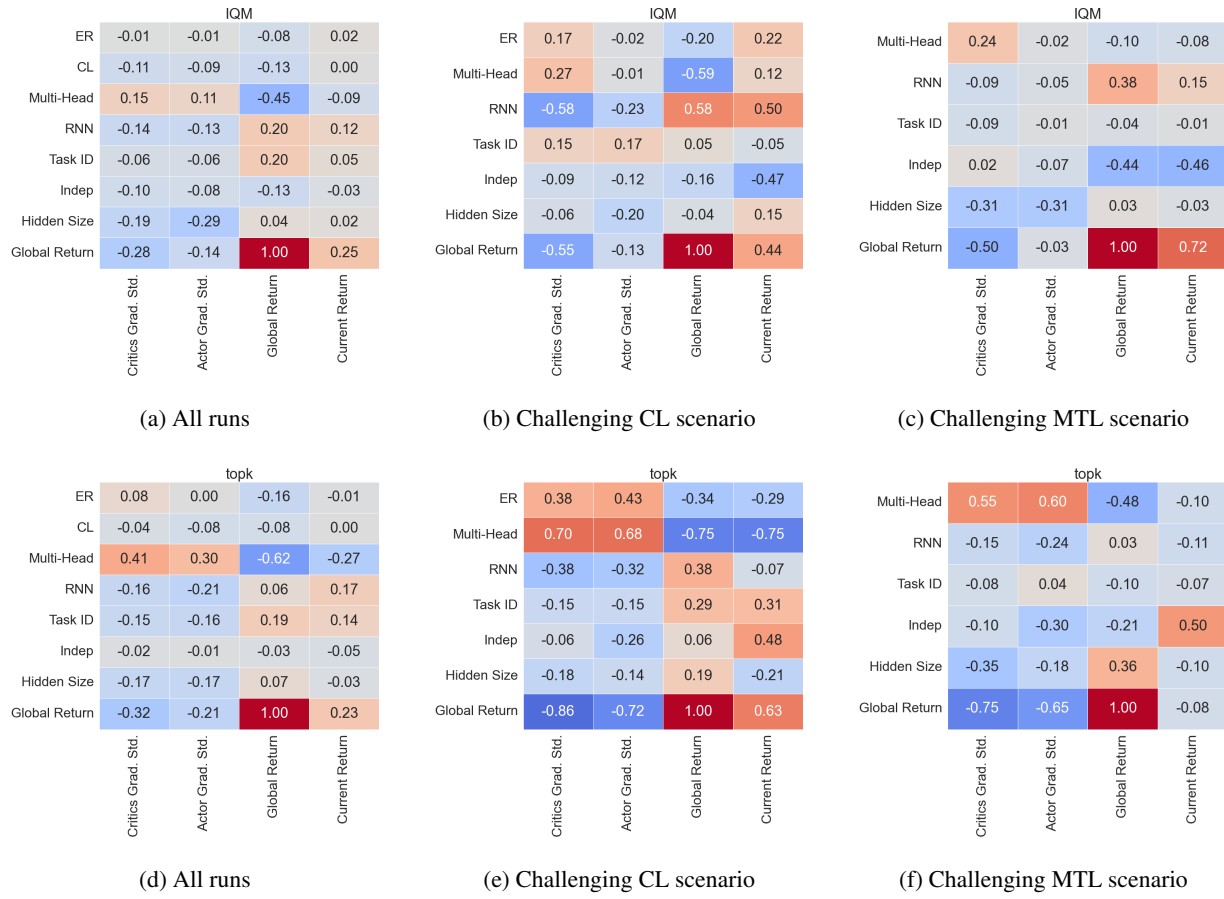

Figure 11: **Spearman correlation matrix for the synthetic experiments (80k runs).** On top are the IQM plots, whereas top 10% ones are at the bottom.

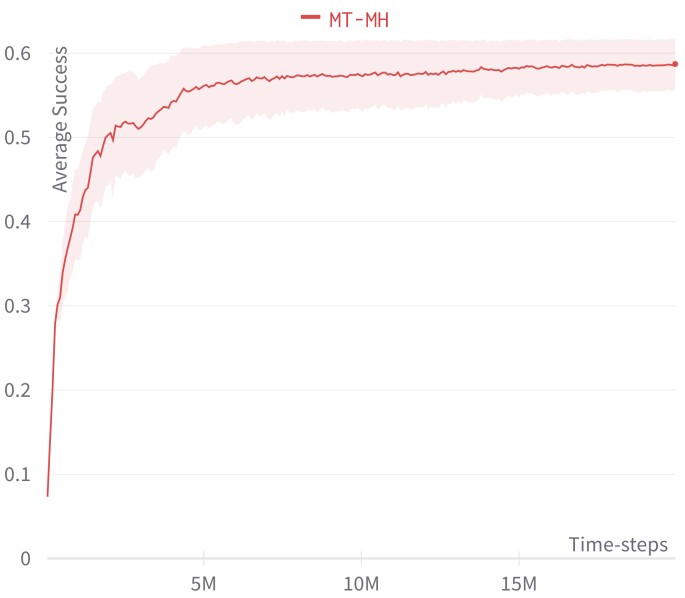

Figure 12: **MT10 experiment**. We repeat the popular MT10 benchmark with our MT-MH implementation. After 20M time-steps, the algorithm reaches a success rate of 58%. This is in line with Meta-World reported results. In Figure 15 of their Appendix, their MT-SAC is trained for 200M time-steps the first 20M time-steps are aligned with our curve. We further note that our CW10 result might seem weak vis à vis the reported ones in Wolczyk et al. (2021b). This is explained by Wolczyk et al. (2021b) using Meta-World-v1 instead of the more recent Meta-World-v2.

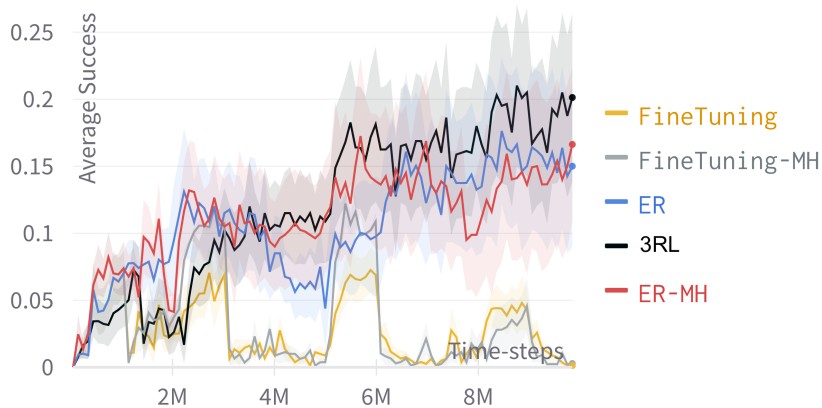

Figure 13: **CW10 experiment with larger neural networks.**. We repeated the CW10 experiment, this time with larger neural networks. Specifically, we added a third layer to the actor and critics. Its size is the same as the previous two, i.e. 400. The extra parameters have hindered the performance of all baselines. Note that it is not impossible that more well-suited hyperparameters could increase the performance of the bigger networks.

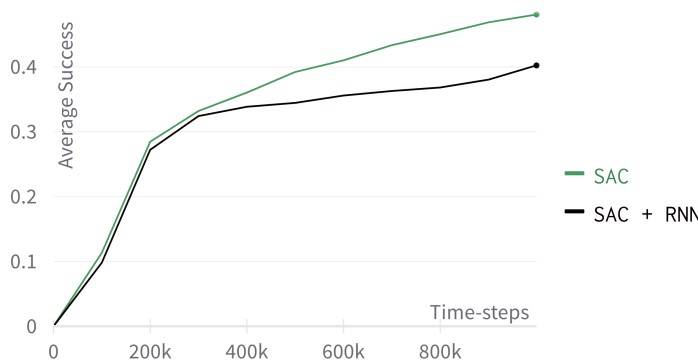

Figure 14: **Single-task learning CW10 experiments**. Average success on all task trained independently. In this regime, the RNN does not help.

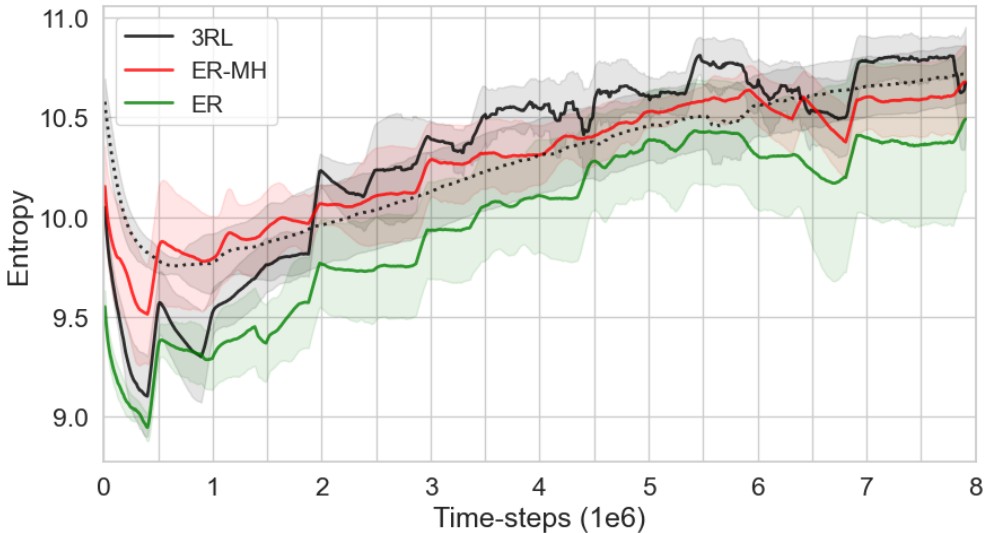

Figure 15: **3RL does not achieve superior continual learning performance through increased parameter stability**. We show the evolution of the methods' entropy in the parameters updates. We include MTL-RNN (dotted line) as a ref. We do not observe an increase in parameter stability: on the contrary, all methods, increasingly update more weights as new tasks (or data) come in.

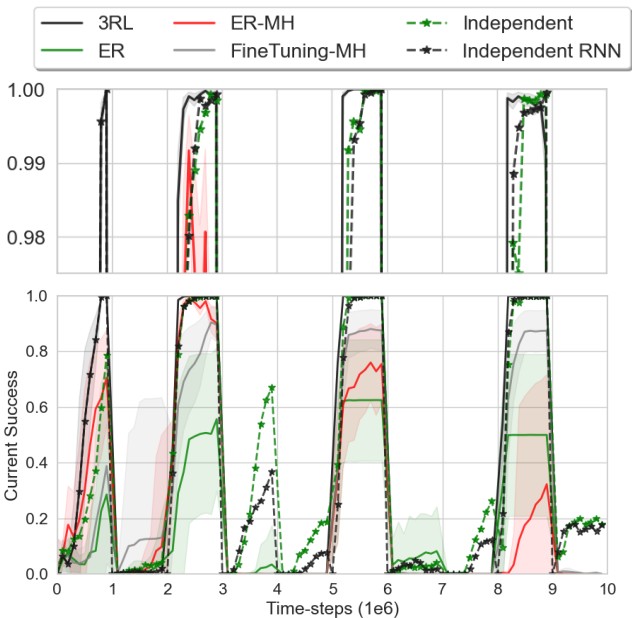

Figure 16: **3RL is the fastest continual learner.** Current success rate on `CW10`. The Independent method, which trains on each task individually, is still the best approach to maximise single-task performance. However, on the task that the continual learning methods succeed at, 3RL is the fastest learner (see the zoom in the top plot). In these cases, its outperformance over Independent and Independent RNN indicates that forward transfer is achieved.

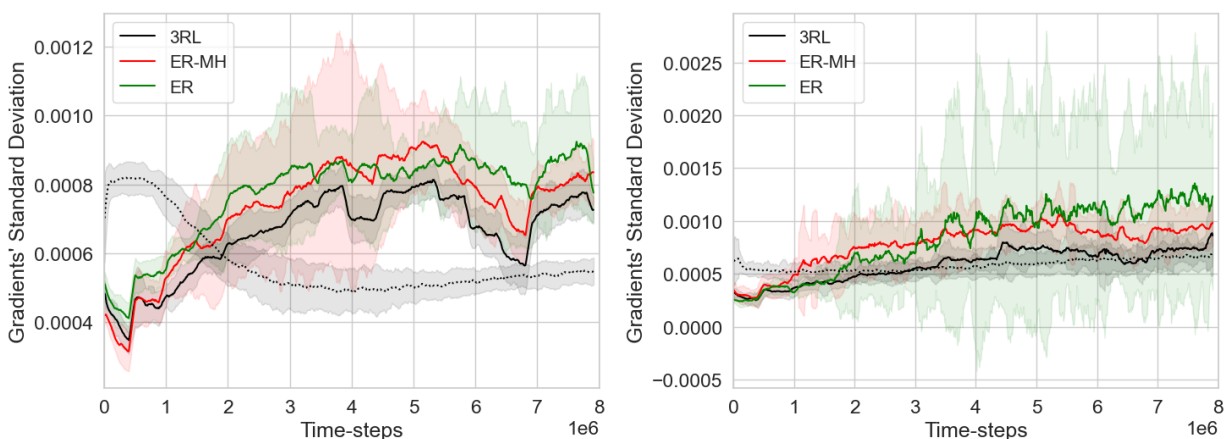

Figure 17: **Gradient variance analysis on CW20**. Comparison of the normalized standard deviation of the gradients for the actor (left) and critics (right) in for different CRL methods. For reference, we included MTL-RNN as the dotted line. The gradient alignment's rank is perfectly correlated with the performance rank.

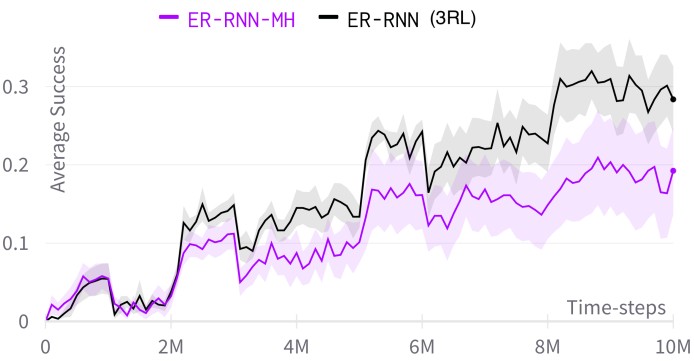

Figure 18: **CW10 experiment combining the RNN and MH**. Combining the best task-agnostic (3RL) and task-aware (ER-MH) CRL methods did not prove useful. Note that this experiment was ran before we enstored gradient clipping, which explains why the performance is lower than previously reported.

