# OpenReview forum: "Task-Agnostic Continual Reinforcement Learning: Gaining Insights and Overcoming Challenges"
_ICLR.cc/2023/Workshop/RRL — RRL 2023 Poster_

### Official Review · Reviewer_Deoj · 2023-02-27

**Rating:** 3
**Confidence:** 3

**Review:**

The paper presents a simple method for continual RL where the task information is not available to the agent. The method combines replay and recurrent mechanisms with a SAC base. The authors find the derived algorithm ‘3RL’ is a strong method across CW10 and MW20.

Strengths:
- Thorough empirical evaluation and rigorous comparison to the immediate baselines.
- Interesting investigation in explaining the forward transfer abilities of 3RL

Weaknesses:
- It is not clear in Algorithm 1 how the buffer D_old is populated.
- Whilst from the point of view of the RL algorithm, the task information is not included, it appears that task identifiers are implicitly still used to split the replay data. Thus, the algorithm is not truly task agnostic, and it would be useful for the authors to comment on this.
- It would be useful to explain what the representation is over in Figure 1 (explained later in Section 3.3)

Minor:
- It is inappropriate to include an identifying repository in a double-blinded review. I would advise the authors link to an anonymous repository in the future.
- Missing Appendix label for Appendix K on page 8 and Appendix H on page 7.

---

### Official Review · Reviewer_wTTE · 2023-03-01
**Review of Paper32**

**Rating:** 4
**Confidence:** 4

**Review:**

This paper produces a clear and well-motivated analysis of the strengths and limitations of continual learning agents. The authors demonstrate how CL agents compare against, multi-task agents, which are trained on all available tasks jointly, and how they are evaluated in the task-agnostic settings, where knowledge of the task ID is unavailable at training time. In general, the authors present two observations that seem to contradict common beliefs in CL; the first being that task-agnostic CL agents with recurrent memory can outperform task-aware agents and the second being that replay-based recurrent reinforcement learning (3RL) agents can reach the performance of its multi-task upper-bound despite being exposed to tasks in a sequential fashion.

Additionally, the authors propose four hypotheses that aim to further explain these results and conduct a large empirical study to compare different RL methods in various multi-task learning regimes. They show that 3RL quickly infers how new task relate to previous ones, enabling forward transfer of knowledge, and learns representations of underling MDPs that reduce task interference.

Overall, the paper is well-written and informative, providing insights into the potential of CL agents paired with recurrent mechanisms.